# Pharmacological Prevention of Ectopic Erythrophagocytosis by Cilostazol Mitigates Ferroptosis in NASH

**DOI:** 10.3390/ijms241612862

**Published:** 2023-08-16

**Authors:** Joon Beom Park, Kangeun Ko, Yang Hyun Baek, Woo Young Kwon, Sunghwan Suh, Song-Hee Han, Yun Hak Kim, Hye Young Kim, Young Hyun Yoo

**Affiliations:** 1Department of Anatomy and Cell Biology, Dong-A University College of Medicine, Busan 49201, Republic of Korea; csplen1990@naver.com (J.B.P.); kku0803@naver.com (K.K.); sewybest@naver.com (W.Y.K.); 2Department of Gastroenterology, Dong-A University College of Medicine, Busan 49201, Republic of Korea; p100100@dau.ac.kr; 3Department of Endocrinology, Dong-A University College of Medicine, Busan 49201, Republic of Korea; suhs@dau.ac.kr; 4Department of Pathology, Dong-A University College of Medicine, Busan 49201, Republic of Korea; freeate@dau.ac.kr; 5Department of Anatomy, School of Medicine, Pusan National University, Yangsan 50612, Republic of Korea; yunhak10510@pusan.ac.kr; 6Department of Anatomy, Inje University College of Medicine, Busan 47392, Republic of Korea

**Keywords:** NASH, cilostazol, hepatic iron overload, ferroptosis, erythrophagocytosis

## Abstract

Hepatic iron overload (HIO) is a hallmark of nonalcoholic fatty liver disease (NAFLD) with a poor prognosis. Recently, the role of hepatic erythrophagocytosis in NAFLD is emerging as a cause of HIO. We undertook various assays using human NAFLD patient pathology samples and an in vivo nonalcoholic steatohepatitis (NASH) mouse model named STAM^TM^. To make the in vitro conditions comparable to those of the in vivo NASH model, red blood cells (RBCs) and platelets were suspended and subjected to metabolic and inflammatory stresses. An insert-coculture system, in which activated THP-1 cells and RBCs are separated from HepG2 cells by a porous membrane, was also employed. Through various analyses in this study, the effect of cilostazol was examined. The NAFLD activity score, including steatosis, ballooning degeneration, inflammation, and fibrosis, was increased in STAM^TM^ mice. Importantly, hemolysis occurred in the serum of STAM^TM^ mice. Although cilostazol did not improve lipid or glucose profiles, it ameliorated hepatic steatosis and inflammation in STAM^TM^ mice. Platelets (PLTs) played an important role in increasing erythrophagocytosis in the NASH liver. Upregulated erythrophagocytosis drives cells into ferroptosis, resulting in liver cell death. Cilostazol inhibited the augmentation of PLT and RBC accumulation. Cilostazol prevented the PLT-induced increase in ectopic erythrophagocytosis in in vivo and in vitro NASH models. Cilostazol attenuated ferroptosis of hepatocytes and phagocytosis of RBCs by THP-1 cells. Augmentation of hepatic erythrophagocytosis by activated platelets in NASH exacerbates HIO. Cilostazol prevents ectopic erythrophagocytosis, mitigating HIO-mediated ferroptosis in NASH models.

## 1. Introduction

Nonalcoholic fatty liver disease (NAFLD) covers a spectrum ranging from nonalcoholic fatty liver (NAFL) to nonalcoholic steatohepatitis (NASH). NAFL shows simple steatosis alone with a benign prognosis in general. However, NASH, characterized by steatosis, ballooning degeneration, inflammation, and fibrosis, is a potentially progressive liver disease that can progress to liver cirrhosis and hepatocellular carcinoma (HCC) [1]. The progression of NAFL to NASH is explained by the “multiple hit” hypothesis. Increased hepatic fat and insulin resistance at the onset of NAFL represent a “first hit” and other “hits” are required for NAFL to progress into NASH [2]. Among these hits, platelets (PLTs) are now known to have novel roles, especially in hepatic inflammation [3], and are considered potent mediators of innate immunity and inflammation [4]. A decrease in PLT count and an increase in mean PLT volume reflect the severity of NASH in patients and are regarded as risk and prognostic factors for NASH. The changes in these values indicate augmentation of PLT activation and PLT consumption in NASH [5,6]. Recently, anti-PLT therapy has been suggested as a promising therapeutic target for NASH and subsequent HCC [7,8].

The roles of PLTs in the pathogenesis of atherosclerosis have been well established. Monocytes recruited to blood vessels differentiate into macrophages and turn into foam cells while phagocytosing low-density lipoprotein. Activated PLTs facilitate monocyte recruitment and introduce an inflammatory milieu. In advanced atherosclerotic plaques, however, intraplaque hemorrhage occurs and augments erythrophagocytosis [9]. The cholesterol content of the RBC membrane leads to lipid deposition in the plaque. Moreover, hemoglobin and its metabolites (heme, iron) possess proinflammatory and pro-oxidative abilities with consequent macrophage ferroptosis, eventually falling into a vicious cycle throughout the progression of atherosclerosis [10]. The roles of macrophages, PLTs, and erythrocytes in NASH are thought to be comparable to their roles in atherosclerosis. Activated PLTs contribute to activating Kupffer cells and progressing NASH to HCC [8]. Activated PLTs also induce phosphatidylserine (PS) exposure on the red blood cell (RBC) membrane with resultant erythrophagocytosis [11]. In physiologic states, erythrophagocytosis and iron recycling mainly occur in the spleen. However, when the requirement for the removal of PS-positive erythrocytes increases, the primary organ of erythrophagocytosis shifts from the spleen to the liver [12]. Increased erythrophagocytosis in the NASH liver causes iron deposition in the liver, with each promoting the other, consequently accelerating ferroptosis and NASH progression [13,14,15].

Cilostazol is an anti-PLT drug that inhibits phosphodiesterase 3A (PDE3A) and phosphodiesterase 3B (PDE3B). PDE3A is mainly expressed in PLTs, cardiac ventricular myocytes, and vascular smooth muscle myocytes, whereas PDE3B is abundant in hepatocytes and adipocytes [16]. PDE3A inhibition by cilostazol blocks the PLT activation cascade by increasing the level of cAMP, which reduces intracellular calcium levels and induces protein kinase A (PKA) activity [17]. Cilostazol also shows lipid-lowering effects in many reports, including in vivo and human studies. Decreased levels of triglyceride, cholesterol, and remnant lipoprotein and increased levels of high-density lipoprotein were observed in the blood, liver, and atherosclerotic plaques [18]. It is thought that elevated PKA levels caused by PDE3B inhibition result in the upregulation of lipoprotein lipase in adipose tissue and the downregulation of lipogenesis in the liver. In addition, PKA-mediated activation of cAMP-response-element-binding-protein (CREB) by cilostazol shows anti-inflammatory effects, including downregulation of NF-κB and TNF-α and upregulation of IL-10. Cilostazol dampens ROS production by suppressing CYP2E1. Additionally, cilostazol has anti-apoptotic effects by inhibiting the activation of BAX and caspase-3 and the formation of the death-inducing signaling complex (DISC) initiated by FasR-FasL binding [16].

In the present study, we observed that PLT and erythrophagocytosis impact NASH progression and that cilostazol prevents ectopic erythrophagocytosis and mitigates hepatic iron overload (HIO)-mediated ferroptosis, alleviating NASH progression.

## 2. Results

### 2.1. NASH Was Induced in STAM^TM^ Mice

NASH was induced in 8-week-old C57BL/6 STAM^TM^ mice. Both liver size and liver weight increased, and macrovesicular steatosis was induced in STAM^TM^ mice (Figure 1A–C). Additionally, ballooning degeneration, inflammation, steatosis, and fibrosis scores were increased in STAM^TM^ mice (Figure 1D). The NAFLD activity score (NAS) developed by Kleiner et al. [19] was measured by the authors and the deep learning module developed by Heinemann et al. [20] and showed that NASs were higher in STAM^TM^ mice. NASs obtained by the deep learning module were relatively lower than those obtained by the authors because the deep learning module treats microvesicular steatosis as an ignored class. Western blot analysis in the liver showed that IL-6 and F4/80 levels were elevated in STAM^TM^ mice, indicating the upregulated recruitment of monocyte-derived macrophages in the NASH liver (Figure 1E). In our study, we utilized a NASH-induced model characterized by type 1 diabetes. This was achieved through a combination of streptozotocin (STZ) administration and a high-fat diet. STZ is known to selectively destroy insulin-producing beta cells in the pancreas, resulting in insulin deficiency and the development of type 1 diabetes. The concurrent feeding of a high-fat diet further exacerbated the metabolic profile, aligning with the model’s characteristics. The blood glucose levels over the entire time course of the GTT and ITT were significantly higher in STAM^TM^ mice (Figure 1F). These results indicate that we appropriately established STAM^TM^ mice as an in vivo NASH model. Interestingly, we observed that a small amount of hemolysis occurred in the serum of STAM^TM^ mice. The quantification of serum hemoglobin, heme, and hemopexin showed that a proportion of RBCs died before blood was drawn (Figure 1G).

### 2.2. Cilostazol Ameliorates Ballooning Degeneration and Inflammation in the NASH Model

Because our previous study demonstrated that cilostazol improves high-fat-diet(HFD)-induced hepatic steatosis [17], we examined whether cilostazol blocks NASH in STAM^TM^ mice (Figure 2A). Cilostazol did not ameliorate hepatomegaly or increase the ratio of liver weight to body weight or the steatosis score in NASH mice (Appendix A). Cilostazol did not improve the changes in lipid or glucose profiles in NASH mice (Appendix A). Nevertheless, STAM^TM^ mice administered cilostazol showed more microvesicular steatosis in place of ballooning degeneration, which could be interpreted as reduced damage (Figure 2B). Furthermore, cilostazol treatment ameliorated NASH as determined by ballooning degeneration and inflammation scores (Figure 2C,D) and the NAS (Figure 2F,G). However, there was no significant difference in the fibrosis score between control mice and mice administered cilostazol (Figure 2E). Western blot analysis showed that cilostazol administration prevented the increase in IL-6 and F4/80 levels observed in NASH mice, indicating that the upregulated recruitment of monocyte-derived macrophages in NASH livers was resolved by cilostazol (Figure 2H).

### 2.3. Cilostazol Resolved the Augmentation of PLT and RBC Accumulation in NASH Livers and Prevented the PLT-Induced Increase in Erythrophagocytosis In Vitro

Because cilostazol did not improve lipid or glucose profiles in NASH mice, we hypothesized that cilostazol exerts a NASH ameliorating effect based on its anti-PLT activity. We examined our hypothesis using human NASH patient pathology samples and in vivo/in vitro NASH models. Immunohistochemistry demonstrated that the hepatic expression of CD61 as a PLT marker was markedly increased in liver tissue from NASH patients compared to liver tissue from simple steatosis patients (Figure 3A) and NASH mice (Figure 3B). In NASH livers, PLTs converged at inflammatory foci and were also distributed around parenchymal cells (Figure 3B). Administration of cilostazol to NASH mice mitigated PLT accumulation in inflammatory foci in the liver (Figure 3B,C). Immunohistochemistry of hemoglobin-α showed that RBCs increased at inflammatory foci and cilostazol resolved the accumulation of RBCs, indicating that cilostazol ameliorated erythrophagocytosis by Kupffer cells in NASH livers (Figure 3D). To simulate erythrophagocytosis by Kupffer cells in NASH liver in vitro, RBCs were cultured with CD68-FITC-labeled THP-1 cells under metabolic and inflammatory stresses. Microplate fluorescence photometry showed that PLTs increased the population of THP-1 cells phagocytosing RBCs irrespective of the presence of metabolic or inflammatory stresses, and cilostazol prevented the increase in the population of THP-1 cells phagocytosing RBCs (Figure 3E). Flow cytometry and confocal microscopy also showed that cilostazol prevented the increase in the population of THP-1 cells phagocytosing RBCs and the generation of a metabolic and inflammatory milieu in a dose-dependent fashion (Figure 3F,G).

### 2.4. Cilostazol Inhibited PLT-RBC Contact-Induced Phosphatidylserine Exposure and Hemolysis

Notably, cilostazol ameliorated hemolysis in STAM^TM^ mice (Figure 4A). The quantification of serum hemoglobin (Figure 4B) and heme (Figure 4C) also showed that cilostazol treatment attenuated hemolysis in NASH mice. We also observed that cilostazol has a palliative effect on RBC viability by inhibiting the inflammatory and metabolic milieu (Figure 4D,E). Therefore, we postulated that cilostazol, as an anti-PLT agent, suppresses PLT activation and consequently rescues the viability of RBCs. PLT activation is known to augment phosphatidylserine exposure of RBCs by binding FasL of PLTs with FasR of RBCs [11]. Our in vitro RBC-PLT interaction study showed that RBC-PLT binding was markedly increased in the inflammatory and metabolic milieu generated by cilostazol. Importantly, cilostazol decreased RBC-PLT binding (Figure 4F). Flow cytometry using PKH26-labeled RBCs and PKH67-labeled PLTs demonstrated that most RBCs bind to PLTs in the inflammatory and metabolic milieu. Notably, cilostazol significantly dampened the ratio of PLT-RBCs to total RBCs in a dose-dependent manner (Figure 4G). RBC-PLT binding induces the Fas pathway in both RBCs and PLTs resulting in phosphatidylserine exposure [21]. Thus, we next examined RBC-PLT binding by Annexin-V flow cytometry. We observed that cilostazol markedly decreased Annexin-V binding of PLTs and RBCs, indicating that cilostazol attenuated the activation of PLTs and apoptosis as well as apoptotic “eat me” signals (Figure 4H,I).

### 2.5. Cilostazol Improved Iron Overload in NASH Livers

We observed that iron overload was induced in NASH livers near inflammatory foci, and cilostazol administration improved HIO (Figure 5A). Next, we examined the expression level of proteins related to hemoglobin-iron metabolism. Western blot assays showed that heme carrier protein 1 (HCP1), localized to the apical membrane for heme transport, was markedly elevated in NASH livers [22], and its level was normalized by cilostazol administration (Figure 5B). Although the expression level of heme-responsive gene 1 (HRG1), crucial to heme transport from the phagolysosome, was not markedly elevated in NASH liver, cilostazol administration markedly decreased its expression level (Figure 5B). A previous study reported that the NASH liver becomes the major site of erythrophagocytosis, surpassing the spleen [12]. We examined whether erythrophagocytosis is shifted from the spleen to the liver in our experimental model. We observed that the levels of heme and iron increased in the NASH liver and that cilostazol reversed this increase (Figure 5C); in the spleen, the opposite pattern was observed (Figure 5D). These findings indicate that upregulated erythrophagocytosis in NASH livers is correlated with HIO and that cilostazol prevents erythrophagocytosis by improving HIO. To confirm this hypothesis in vitro, we adopted an insert-coculture system in which activated THP-1 cells and RBCs were seeded on the inside plate and HepG2 cells were seeded on the outside plate, with the two plates separated by a 0.4 μm porous membrane (Figure 5E). The protein level of HCP1 was elevated in both THP-1 and HepG2 cells in the presence of RBCs and was diminished with cilostazol treatment (Figure 5F). The level of HRG1 in THP-1 cells was elevated in the presence of RBCs and diminished with cilostazol treatment (Figure 5G). However, the level of HRG1 in HepG2 cells was not elevated in the presence of RBCs (Figure 5G), supporting our western blot data in which the total expression level of HRG1 was not increased in NASH livers.

### 2.6. Cilostazol Attenuated RBC-Induced Ferroptosis of Hepatocytes and Phagocytes

Using an insert-coculture system, we dissected the cell death mode. Treatment with RBCs reduced the viability of THP-1 and HepG2 cells (Figure 6A), and the ratio of cleaved caspase-3 to procaspase-3 in THP-1 and HepG2 cells (Figure 6B), indicating that programmed cell death was induced. Cilostazol reversed RBC-induced cell death in a dose-dependent manner (Figure 6A,B). Both apoptosis-inducing factor mitochondria-associated 2 (AIFM2) and glutathione peroxidase 4 (GPX4) have anti-lipid peroxidation activity that can protect cells from ferroptosis and their level increases upon ferroptosis [23]. Notably, treatment with RBCs increased the protein expression levels of AIFM2 and GPX4 in THP-1 and HepG2 cells and cilostazol attenuated the increased expression of these two proteins (Figure 6C,D). To examine whether ferroptosis underlies erythrophagocytosis-induced abnormal iron overload in the NASH liver, we undertook immunohistochemistry using human NASH patient pathology samples. Immunohistochemistry demonstrated that the protein expression of AIFM2 was markedly increased in liver tissue from NASH patients compared to that in liver tissue from simple steatosis patients (Figure 6E). Flow cytometry and confocal microscopy demonstrated that cilostazol mitigated this lipid peroxidation in THP-1 and HepG2 cells in a dose-dependent manner (Figure 6F,G). While the levels of lipid peroxidation in HepG2 cells showed a normal distribution, there were two populations of THP-1 cells: a population sensitive to lipid peroxidation and another population resistant to lipid peroxidation. Treatment with cilostazol reduced the proportion of the THP-1 cell population sensitive to lipid peroxidation (Figure 6F).

Taken together, hepatocytes and phagocytes undergo cell death at least partly through hepatic erythrophagocytosis-induced ferroptosis in NASH liver. Presumably, the efficacy of cilostazol in alleviating NASH depends on the relief of hepatic erythrophagocytosis in phagocytes and subsequent ferroptosis of both phagocytes and hepatocytes.

### 2.7. Our Findings Were Validated through Gene Expression Omnibus (GEO) Data Analysis

The overall hypothesis of “PLC activation—RBC fragility—intrahepatic erythrophagocytosis—heme induced iron overload—ferroptosis” was evaluated through GEO data analysis of human NASH liver, rat NASH liver, and mouse NASH liver. Three GEO datasets (GSE126848, GSE129525, and GSE156918) were downloaded and processed according to classification by GO terms involved in the catabolism and anabolism of hemoglobin, heme and iron, and ferroptosis. A total of 835 genes of Homo sapiens related to iron metabolism were analyzed in the datasets. The value of each gene, converted to a log-ratio to its mean, was expressed as a heatmap plot in which green represents downregulated genes and red represents upregulated genes (Figure 7A). Iron metabolism-related genes were elevated in NAFLD and NASH. Interestingly, however, almost all genes involved in the metabolism of hemoglobin and heme and ferroptosis were upregulated in NAFLD and NASH in humans, rats, and mice. In the volcano plot, the *p*-value of most genes was below 0.05, indicating statistical significance in humans, rats, and mice (Figure 7B–D). In the GO enrichment analysis, nearly all pathways were significantly upregulated (Figure 7E). Interpretation of these data is limited in that they were obtained from whole tissue lysates of the liver. For example, upregulated gene expression associated with iron export does not imply alleviation of hepatic iron overload because it could be the result of increasing iron exchange between hepatic parenchymal cells. Further studies, such as single-cell transcriptomics, are needed.

## 3. Discussion

The present study demonstrated that cilostazol has potential therapeutic effects in ameliorating the pathological progression of NASH. The STAM^TM^ NASH mice model used in this study offers the advantage of monitoring the natural progression from simple hepatic steatosis to NASH and fibrosis in a relatively controllable manner. It defines the onset of NASH and fibrosis as occurring between 6–8 weeks and 9–12 weeks respectively. We previously illustrated that cilostazol ameliorates HFD-induced hepatic steatosis by lowering blood lipids through an energy sensor, AMPK [17]. However cilostazol ameliorated ballooning degeneration and inflammation in NASH livers, it did not improve lipid or glucose profiles. These findings suggest that the efficacy of cilostazol in the amelioration of NASH appears by different mechanisms from previous studies.

There is accumulating evidence for the involvement of hepatic (programmed) cell death including apoptosis and necroptosis in the pathogenesis of NASH. Thus, the cytokeratin 18 (CK18) fragment and receptor-interacting serine/threonine-protein kinase 3 (RIPK3) or mixed lineage kinase domain-like protein (MLKL) are often used as the diagnostic biomarkers in NASH because CK18 is released from apoptotic hepatocytes and RIPK3 and MLKL, are upregulated in necroptosis [24,25]. On the other hand, ferroptosis is a recently identified programmed necrosis, which is characterized by excess iron accumulation and lipid peroxidation [23]. Peroxidation of accumulated lipids may play an important role in the pathogenesis of NASH because it is considered to trigger hepatocellular injury, hepatic fibrogenesis, and the carcinogenesis of hepatocytes. In contrast to strictly controlled apoptosis, cells undergoing ferroptosis release proinflammatory factors, such as damage-associated molecular pattern molecules called DAMPs [26]. The released DAMPs damage neighboring cells and eventually induce ferroptosis, generating a vicious cycle. In coculture experiments with HepG2 and THP-1 cells, GPX4 and AIFM2, which are markers of ferroptosis, were upregulated when THP-1 cells phagocytosed RBCs. Our findings indicate that ferroptosis underlies hepatic damage induction in NASH. Another important marker of ferroptosis, lipid peroxide, was also exacerbated in NASH in vitro.

HIO is an emerging issue commonly found in NAFLD and an exacerbating factor of NAFLD itself [14,15]. HIO is a frequent finding in the NAFLD population and is an aggravating factor for NAFLD. Some groups even suggest phlebotomy as a treatment option for NAFLD with HIO [27]. We previously reported that iron accumulates in NAFLD patients and in in vitro and in vivo NAFLD models and that recombinant fibroblast growth factor 21 (FGF21) ameliorates NAFLD by attenuating HIO [14]. In the present study, we observed an increase in hepatic iron input through upregulated erythrophagocytosis in NASH livers, indicating that increased erythrophagocytosis is a major mechanism of HIO in NASH. Iron overload in the NASH liver is an aggravating factor for NASH itself because iron acts as a generator of ROS, which is a major pathogen of NASH. To date, the origin of iron in NASH liver remains unclear, but our findings suggest that hepatic erythrophagocytosis may be a significant iron source in the livers of patients with NASH. Cilostazol seems to ameliorate NASH by reducing HIO.

Erythrophagocytosis in NAFLD liver is thought to be increased by activation of Kupffer cells and reduction of RBC lifespan. Recent studies of NAFLD/NASH highlight the roles of PLTs in provoking and aggravating NAFLD. The known roles of PLTs in NAFLD are mostly associated with proinflammatory effects in the liver, which may eventually lead to NASH-cirrhosis-HCC progression [3]. A report showed that PLTs activate membrane-bound FasL to induce apoptosis in other cells [28]. Another study reported that phosphatidylserine externalization at the RBC membrane is the result of the PLT-RBC interaction caused by FasL-FasR binding and that gene deletion of either FasL or FasR reduces phosphatidylserine exposure of RBCs and PLTs [11]. Overall, the data suggest that the role of PLTs in inducing HIO is the result of erythrophagocytosis induction in the liver and proinflammatory effects. We, in the present study, observed PLT-RBC binding and hepatic iron accumulation in NASH models.

Cilostazol is an anti-PLT drug. There is a report suggesting that aspirin and clopidogrel did not suppress PLT-mediated induction of phosphatidylserine exposure in RBCs [11]. In the present study, we found that cilostazol inhibited the interaction between PLTs and RBCs and consequently inhibited phosphatidylserine exposure in RBCs. Although RBCs can undergo apoptosis in certain circumstances, such as sepsis and malaria, this pathway is limited because RBCs are anucleate cells [29,30]. The disposal of RBCs largely depends on erythrophagocytosis by splenic macrophages. However, when the demand for RBC disposal is excessive, hepatic macrophages perform erythrophagocytosis in the liver [12]. We found that hepatic erythrophagocytosis is exacerbated by activated PLTs, and cilostazol treatment alleviated this hepatic erythrophagocytosis in NASH, which mainly depends on the inhibition of PLT-mediated phosphatidylserine exposure of the RBC membrane.

Taken together, the NASH liver seems to fall into a vicious cycle through the following successive mechanisms: PLT activation—RBC fragility—intrahepatic erythrophagocytosis—heme-induced iron overload—ferroptosis. Cilostazol has therapeutic potential through the suppression of this vicious cycle. We further evaluated liver tissue to examine the effect of cilostazol on the progression of NASH, focusing on tumorigenesis. Fourteen-week-old STAM^TM^ mice showed a well-circumscribed mass composed of cells with nuclear atypia forming the trabecular pattern, histologically consistent with well-differentiated hepatocellular carcinoma. In contrast, cilostazol-administered 14-week-old STAM^TM^ mice exhibited no definite mass-like lesion (Appendix A). These results support our hypothesis that cilostazol prevents progression to HCC by inhibiting NASH. Further studies are needed to explore the molecular pathway of cilostazol’s effects on erythrophagocytosis.

In conclusion, this study elucidates the pivotal role of cilostazol in understanding and treating the complex pathological process of NASH. We have identified that cilostazol mitigates NASH through a sequential mechanism involving the suppression of PLT activation, RBC fragility, intrahepatic erythrophagocytosis, heme-induced iron overload, and ferroptosis. These findings suggest a significant therapeutic potential for cilostazol in the treatment strategy for NASH patients, particularly focusing on the pathology related to iron overload. The results contribute to the development of therapeutic approaches targeting the intricate interplay of these factors. However, it must be noted that this study was conducted in specific animal models, and further research is required to broaden our understanding of cilostazol’s effects and apply them in a clinical context. Specifically, a deeper understanding of cilostazol’s long-term effects, safety, and interactions with other NASH-related factors is imperative. The insights gained from this study pave the way for future investigations into the molecular pathways of cilostazol’s effects on erythrophagocytosis, offering a promising avenue for the prevention and management of NASH and potentially halting its progression to more severe liver conditions such as HCC.

## 4. Materials and Methods

### 4.1. Human Pathology Samples

This study included 5 unrelated subjects who had undergone a liver biopsy at the Dong-A University Medical Center (Busan, Republic of Korea). Five subjects were proven to have NAFLD histopathologically. NAFLD patients consisted of two cases of simple steatosis and three NASH cases. Patients with secondary causes of steatosis, including alcohol abuse, liver diseases other than NAFLD, total parenteral nutrition, and the use of drugs known to precipitate steatosis, were excluded. The study was approved by the Institutional Review Board of Dong–A University Medical Center (2-1040709-AB-N-01-202202-BR-001-02).

### 4.2. Animal Model

Male C57BL/6J mice were purchased from Samtako, Inc. (Osan, Republic of Korea). The NASH model named STAM^TM^ was induced according to the method used by Fujii et al. [19]. and consisted of a single subcutaneous injection of 200 µg streptozotocin on the second day after birth and feeding with a high-fat diet (5.1 kcal/g, with 60% of calories from fat, 20% from protein and 20% from carbohydrate) starting from 4 weeks of age. The cilostazol group was fed a high-fat diet containing 0.1% cilostazol (Pletaal, Korea Otsuka Pharmaceutical Co., Seoul, Republic of Korea) ad libitum at approximately 100 mg/kg per day. Mice were sampled at 12 weeks of age. All mouse procedures were approved by the Committee on Animal Investigation at Dong-A University (DIACUC-18-7).

### 4.3. Analysis of Lipid Metabolites

Plasma total cholesterol (TC), triglyceride (TG), and nonesterified fatty acids (NEFA) were measured using an enzymatic, colorimetric test kit (Asan Pharmaceutical Co., Seoul, Republic of Korea).

### 4.4. Intraperitoneal Glucose Tolerance Tests (GTTs) and Insulin Tolerance Tests (ITTs)

Mice were fasted for 6 h. Plasma glucose concentrations were measured in tail blood using a GlucoDr Blood Glucose Test Strip (Hasuco, Seoul, Republic of Korea) prior to and 15, 30, 60, 90, and 120 min after intraperitoneally injecting a bolus of glucose (3 mg/g) for the GTT and at the same time points after intraperitoneally injecting 0.75 U/kg body weight insulin for the ITT.

### 4.5. Histology

Liver tissues were fixed in 4% paraformaldehyde, embedded in paraffin, and sectioned to a thickness of 4 μm. Microscopic analysis of whole slides was performed using a Pannoramic MIDI-II Scanner (3D Histech, Budapest, Hungary). Trichrome staining (Polysciences, Inc., Warrington, PA, USA, #25088-1) was scored by the authors according to Kleiner’s NAS system and by a deep-learning module established by Heinemann et al. [19,20] High magnification tiles (299 × 299 px^2^ at 0.44 μm/px) were obtained to analyze ballooning degeneration, inflammation and steatosis, and low magnification tiles (299 × 299 px^2^ at 1.32 μm/px) were generated to analyze fibrosis.

### 4.6. Immunohistochemistry

Immunohistochemistry was performed with primary antibody at 4 °C overnight and then with a matching biotinylated secondary antibody for 1 h at RT. 3,3′-diaminobenzidine (Sigma, St. Louis, MO, USA) was used as chromogen for 2–5 min with hydrogen peroxide substrate following counterstain with hematoxylin. The antibodies used in the above experiment are against anti-hemoglobin α (Abcam, Cambridge, MA, USA, ab92492), and CD61 (Cell signaling, Danvers, MA, USA, #13166) was performed manufacturer’s instruction.

### 4.7. Preparation of RBCs and PLTs

Fresh blood from healthy volunteers was collected in acid citrate dextrose tubes (ACD-A, BD Vacutainer^®^, Becton, Dickinson and Company, Franklin Lakes, NJ, USA) and centrifuged at 900× *g* and 16 °C for 5 min. All procedures with blood sampling were approved by the Institutional Review Board of Dong-A University (2-1040709-AB-N-01-202202-BR-001-02). The lower 1/3rd volume of plasma was centrifuged at 1000× *g* and 16 °C for 10 min. The platelet pellet was suspended in Ca^2+^-free PBS. RBCs were obtained from the red pellet following the first centrifugation and washed 3 times with Ca^2+^-free PBS. Senescent RBCs were prepared by incubating RBCs at 42 °C for 2 h. RBCs and PLTs were stored at 4 °C.

### 4.8. Cell Culture

THP-1 human monocytic cells obtained from Korean Cell Line Bank were maintained in Roswell Park Memorial Institute medium (RPMI 1640, Gibco, Gaithersburg, MD, USA) containing 10% (*v*/*v*) heat-inactivated FBS, 50 mg/mL streptomycin, 50 U/mL penicillin (Gibco) and 0.05 uM 2-mercaptoethanol (Sigma) at 37 °C in a humid atmosphere of 5% CO_2_. THP-1 cells were differentiated to M1 phenotype with 150 nM PMA for 24 h and subsequently incubated with 100 ng/mL LPS and 2 ng/mL INF-γ for 24 h. HepG2 cells obtained from the American Type Culture Collection were maintained in Dulbecco’s Modified Eagle’s Medium with 10% (*v*/*v*) heat-inactivated FBS, 50 mg/mL streptomycin, 50 U/mL penicillin (Gibco) at 37 °C in a humid atmosphere of 5% CO_2_. SPLInsert^TM^ hanging type was used to co-culture HepG2 and THP-1.

### 4.9. Coculture System

To create a coculture system of HepG2 and THP-1 cells, we used the SPLInsert^TM^ hanging type. Two plates were separated with a 0.4 μm porous membrane. HepG2 cells were seeded in 6-well plates, and THP-1 cells were seeded and differentiated in the inner plate, which was also a 6-well plate. THP-1 cells were incubated with 150 nM PMA for 24 h to induce differentiation into the M1 phenotype and subsequently incubated with 100 ng/mL LPS and 2 ng/mL INF-γ for 24 h. After the differentiation of THP-1 cells, the inner plate of THP-1 cells was relocated to a 6-well plate of HepG2 cells. The medium was replaced with fresh medium containing 0.2% (*w/v*) BSA-conjugated 500 μM palmitate, 100 ng/mL LPS, 2 ng/mL INF-γ, and cilostazol. Then, RBCs were treated only on the inner plate.

### 4.10. Hemolysis Analysis

Blood was gently collected with an 18G needle syringe and stored at room temperature for 1 h; then, serum was isolated by centrifugation at 1000× *g* for 10 min. To evaluate hemolysis quantitatively, spectrophotometric absorbance of the serum at a wavelength of 450 nm, which indicates the amount of hemoglobin in the serum, was measured. Heme assays (Mybiosource, San Diego, CA, USA, #MBS841599) and analysis of the level of the heme scavenger protein hemopexin (Mybiosource, #MBS2704013) were performed according to the manufacturer’s instructions.

### 4.11. In Vitro Simulation of NASH and RBC-PLT Interaction Assay

To make RBCs senescent, RBCs were incubated at 42 °C for 2 h. To make comparable in vitro conditions to the in vivo NASH model, RBCs, and PLTs were suspended in Ca^2+^-free PBS with or without cilostazol and were subjected to metabolic stresses (0.2 mM palmitate and 500 mg/dL glucose) and inflammatory stresses (100 ng/mL LPS and 2 ng/mL INF-γ). Then, 10% (*v*/*v*) RBC suspension and 3% (*v*/*v*) platelet-rich plasma were added, and the cells were incubated for 6 h. The absorbance of the supernatant at 450 nm was measured to evaluate hemolysis. RBCs and RBC-bound PLTs were separated from PLTs by centrifugation at 1000× *g* for 10 min with ^Histopaque®^-1077 media.

### 4.12. Erythrophagocytosis Assay

For the PLT-RBC interaction assay, RBCs and PLTs were labeled with PKH26 and PKH67 dyes (Sigma), respectively. For the erythrophagocytosis assay, RBCs were labeled as described above. THP-1 cells were incubated with CD68 (Cell Signaling, #76437) primary antibodies and then with goat anti-rabbit IgG Alexa Fluor^®^ 488 conjugate (Invitrogen) secondary antibodies. BODIPY^TM^ 581/591 C11 (Invitrogen, Carlsbad, CA, USA) was used to detect ROS in HepG2 and THP-1 cells with Hoechst 33342 counterstaining (Invitrogen). Flow cytometry was analyzed with Attune^TM^ NxT Cytometer (Invitrogen) equipped with a 488 nm excitation laser—530/30 nm emission filter and a 561 nm excitation laser—585/16 nm emission filter. Data from the Attune^TM^ NxT Cytometer were acquired and visualized by Attune^TM^ CytPix^TM^ flow cytometers (Invitrogen). RBC and PLT contact was presented as the percentage of RBCs bound to PLTs to total RBCs. Fluorescence images were observed using an LSM 800 confocal microscope (Zeiss, Gottingen, Germany). For microplate fluorescence photometry, THP-1 cells seeded on plates were differentiated by 24 h of incubation with 150 nM PMA and 24 h of subsequent incubation with 100 ng/mL LPS and 2 ng/mL INF-γ. The medium containing RBCs, PLTs, 0.2% (*w*/*v*) BSA-conjugated 500 μM palmitate, 100 ng/mL LPS, and 2 ng/mL INF-γ was then replaced. The cell ratio of THP-1 cells/RBCs/PLTs was 1:10:10. After incubation for 1 h, the plates were washed and analyzed. A SpectraMax^®^ M microplate reader (Molecular Devices, San Jose, CA, USA) was used to measure fluorescence intensity in 96-well plates. Except for 96-well-plate fluorescence, erythrophagocytosis assays were performed without platelets due to clotting control difficulties.

### 4.13. Ferroptosis Assay

Cell viability was determined by a trypan blue exclusion assay. The amount of cell death was quantified by the ratio of cleaved caspase-3 to procaspase-3. The expression levels of GPX4 and AIFM2, which have anti-lipid peroxidation activity and block cells from ferroptosis, were analyzed by western blot assay. To evaluate lipid peroxidation, a key feature of ferroptosis, HepG2 and THP-1 cells were stained with BODIPY^TM^ 581/591 C11 (Invitrogen) to detect ROS with Hoechst 33342 counterstaining (Invitrogen). Flow cytometry and confocal microscopy were undertaken.

### 4.14. Gene Expression Omnibus (GEO) Data Collection and Analysis

We investigated mRNA expression in NAFLD liver tissue from GEO datasets using the NCBI Platform. A total of three original datasets were downloaded (GEO accession no. GSE126848 [31], GSE129525 [32], and GSE156918 [33] arrayed from Homo sapiens, Rattus norvegicus and Mus musculus, respectively) in the format of a tab-separated values file. These datasets were selected based on the following criteria: (1) inclusion of a healthy control group and a NASH group; (2) exclusion of an underlying unrelated disease; (3) exclusion of genetically modified samples; and (4) inclusion of at least six samples per group. The 13 enriched gene ontology (GO) terms in the category of biological processes were all involved in the catabolism and anabolism of hemoglobin, heme iron, and ferroptosis. GSE126848 includes 4 groups: normal (n = 14), obesity (n = 12), NAFLD (n = 15), and NASH (n = 16). GSE129525 includes 3 groups: control (n = 6), NAFLD (n = 6), and NASH (n = 6). GSE156918 includes 2 groups: control (n = 5) and NASH (n = 5). The processing of gene expression data was conducted with Excel in Microsoft Office 365.

### 4.15. Statistical Analysis

There were 8 samples per group in the in vivo experiments. The statistical significance of the differences was determined using the Mann–Whitney U test. Three independent experiments were performed in triplicate in vitro. The results are expressed as the mean ± S.D. obtained from three experiments. The Shapiro–Wilk test was conducted to check the normality of the data, and Levene’s test was used to verify the homogeneity of variances before one-way analysis of variance (ANOVA). ANOVA followed by Scheffe’s test was used for the analysis of differences between each treated condition. *p* < 0.01 was considered statistically significant.

## Figures and Tables

**Figure 1 ijms-24-12862-f001:**
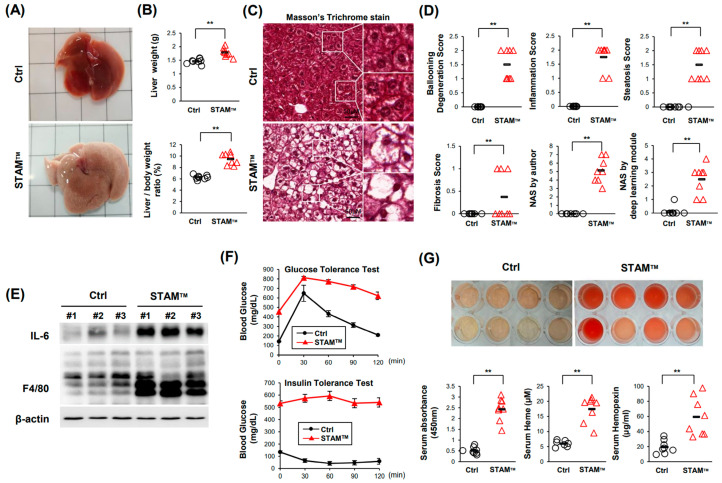
NASH was induced in the STAM^TM^ model. (**A**) The gross phenotype of the liver. (**B**) Liver weight (upper) and liver weight/body weight (lower). (**C**) Masson’s trichrome staining of liver tissue. (**D**) Scores indicating NAFLD activity. (**E**) Western blot data showing the expression levels of the inflammatory factors IL-6 and F4/80. (**F**) Assays showing that insulin resistance was induced in the STAM^TM^ model. GTT (**upper**) and ITT (**lower**). (**G**) Assays showing that hemolysis occurred in the serum of NASH mice. Photograph of serum showing hemolysis, quantification of hemoglobin, heme assay, and hemopexin ELISA. ** *p* < 0.01, Mann–Whitney U test.

**Figure 2 ijms-24-12862-f002:**
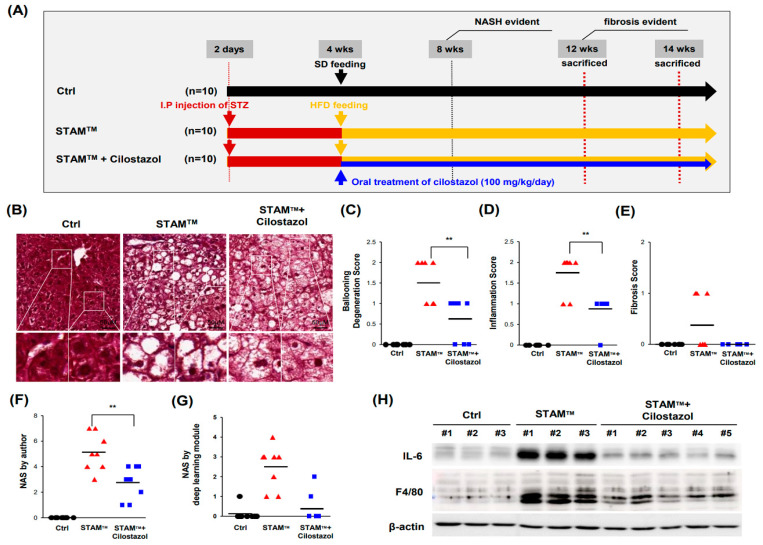
Cilostazol ameliorates ballooning degeneration and inflammation in the STAM^TM^ model. (**A**) Design of the NASH studies using the STAM^TM^ model. (**B**) Masson’s trichrome staining of liver tissue. (**C**–**G**) Score according to Kleiner’s NAS system. (**C**) Ballooning degeneration score. (**D**) Inflammation score. (**E**) Fibrosis score. (**F**) Author-scored NAS value; sum of ballooning degeneration score, inflammation score, steatosis score, and fibrosis score. (**G**) NAS value determined by the deep learning module developed by Heinemann et al. (**H**) Western blot data showing that inflammatory factors (IL-6 and F4/80) were reversed in the cilostazol group. ** *p* < 0.01, Mann–Whitney U test.

**Figure 3 ijms-24-12862-f003:**
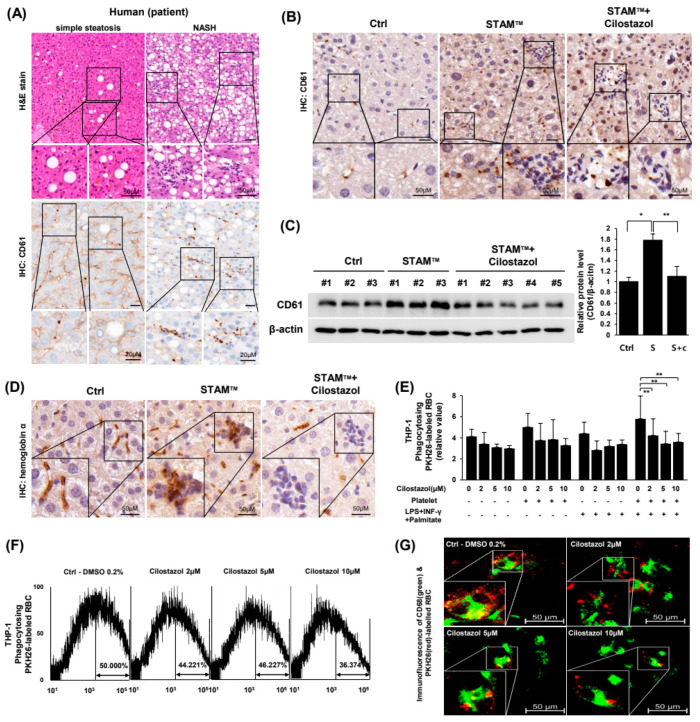
Cilostazol resolved the augmentation of PLT and RBC accumulation in NASH livers and prevented the PLT-induced increase in erythrophagocytosis in vitro. (**A**) H&E staining (upper) and immunohistochemistry of CD61 (lower) in liver tissue from human NASH patients. (**B**) Immunohistochemistry and (**C**) Western blotting data of CD61 in liver tissue from the STAM^TM^ model. * *p* < 0.05 and ** *p* < 0.01. (**D**) Immunohistochemistry of hemoglobin alpha in liver tissue from the STAM^TM^ model. (**E**–**G**) Detection of CD68-FITC-labeled THP-1 cells phagocytosing PKH26-labeled RBCs. (**E**) Microplate fluorescence photometry. * *p* < 0.05 and ** *p* < 0.01. (**F**) Flow cytometry and (**G**) confocal microscopy. Green: CD68, Red: PKH26.

**Figure 4 ijms-24-12862-f004:**
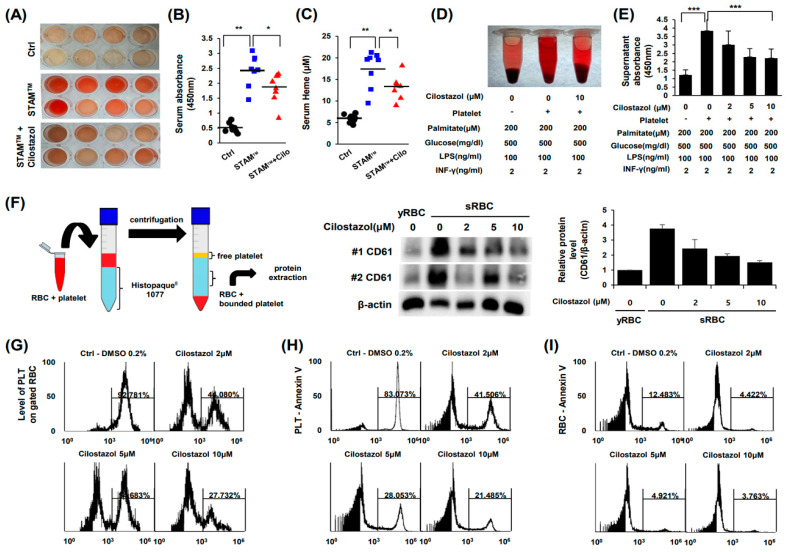
Cilostazol inhibited PLT-RBC contact-induced phosphatidylserine exposure and hemolysis. (**A**–**D**) Analysis of serum from STAM^TM^ mice. (**A**) Photograph of serum. (**B**) Quantification of hemoglobin. (**C**) Heme assay. (**D**,**E**) In vitro simulation of NASH and RBC-PLT interaction assay. (**D**) Photograph of suspension. (**E**) Quantification of hemoglobin. The mean ± S.D. of three independent experiments performed in triplicate is presented. *** *p* < 0.01 according to Scheffe’s test. (**F**) Schematic diagram of the experiment (**left**) and western blotting data (**right**). (**G**) RBC-PLT binding. (**H**,**I**) Flow cytometry using PKH26-labeled RBCs and PKH67-labeled PLTs. (**H**) PLT-Annexin-V. (**I**) RBC-Annexin-V. * *p* < 0.05 and ** *p* < 0.01, Mann–Whitney U test. The values in (**E**) represent the mean ± S.D. of three independent experiments performed in triplicate. *** *p* < 0.01 according to Scheffe’s test.

**Figure 5 ijms-24-12862-f005:**
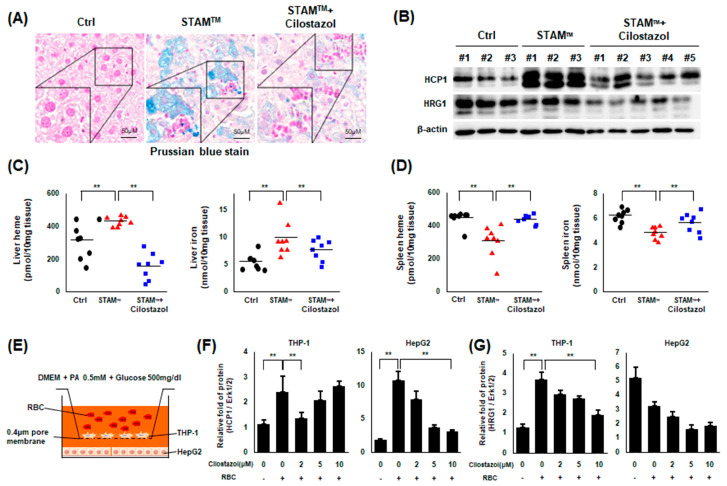
Cilostazol rehabilitated the upregulation of erythrophagocytosis-induced HIO in NASH models. (**A**) Prussian blue staining of liver tissue from the STAM^TM^ model. (**B**) Western blotting data of liver tissue from the STAM^TM^ model. (**C**) Heme and iron assay using liver tissue lysate. (**D**) Heme and iron assay using spleen tissue lysate. Data in (**C**,**D**). ** *p* < 0.01, Mann–Whitney U test. (**E**–**G**) Insert coculture study. (**E**) Schematic illustration. (**F**,**G**) Western blotting data showing the ratio of the corresponding protein to Erk1/2. ** *p* < 0.01 according to Scheffe’s test.

**Figure 6 ijms-24-12862-f006:**
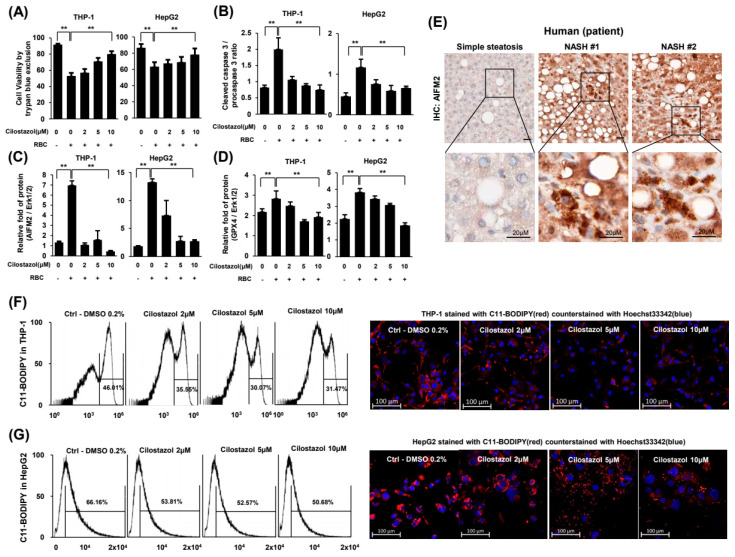
Cilostazol attenuated RBC-induced ferroptosis in THP-1 and HepG2 cells. (**A**–**D**) Coculture experimental system. (**A**) Cell viability assay. (**B**) Quantification of the ratio of cleaved caspase-3 to procaspase-3. (**C**,**D**) Quantification of western blotting for the ratio of AIFM2 (**C**) and GPX4 (**D**) to the loading control. (**E**) Immunohistochemistry of AIFM2 in liver tissue from human NASH patients. (**F**,**G**) Flow cytometry assay and confocal microscopy of THP-1 (**F**) and HepG2 (**G**) cells stained with BODIPY^TM^ 581/591 C11. Confocal microscopy stained with BODIPY^TM^ 581/591 C11 (red) and counterstained with Hoechst 33342 (blue). ** *p* < 0.01 according to Scheffe’s test.

**Figure 7 ijms-24-12862-f007:**
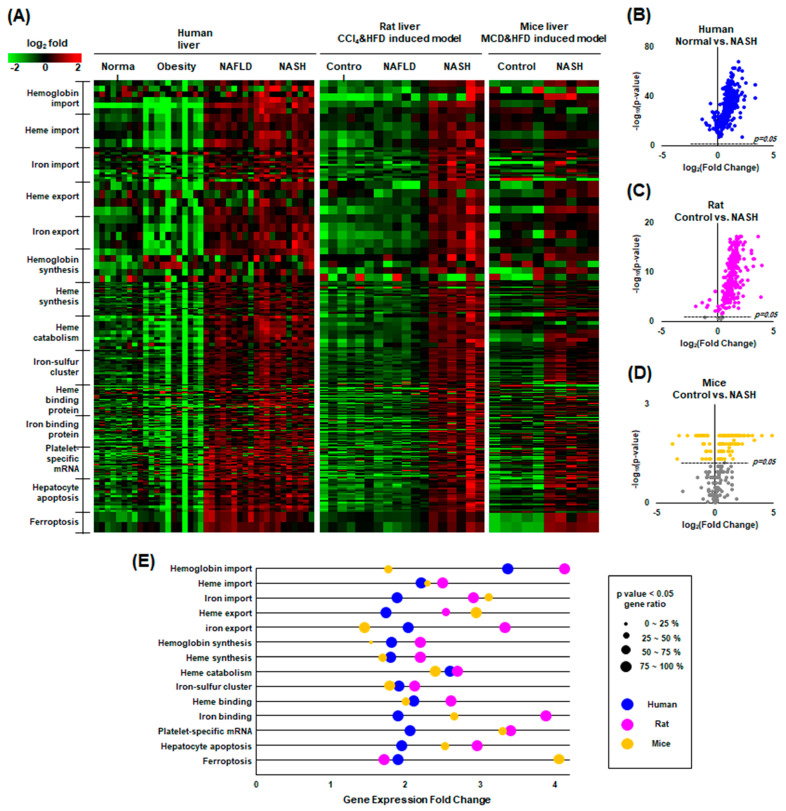
GEO data analysis of human, rat, and mouse NASH. (**A**) Heatmap plot of the factors classified by GO terms related to heme and iron metabolism, showing upward mRNA expression of most factors in NAFLD compared to healthy controls in humans, rats, and mice. Green indicates an above-average increase, red indicates a below-average decrease, and black indicates no difference from the average. The intensity of a color is proportional to the base 2 logarithm. (**B**–**D**) Volcano plots showing that most factors related to heme and iron metabolism are statistically significant. Gray dots are not significant. (**E**) Gene ontology enrichment plot comparing NASH to normal controls.

## Data Availability

Not applicable.

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
