# Peer review of "Pharmacological Prevention of Ectopic Erythrophagocytosis by Cilostazol Mitigates Ferroptosis in NASH"

_ijms, 2023, doi:10.3390/ijms241612862_

Round 1
Reviewer 1 Report
The authors studied the mechanism of alleviation of NASH by Cilostazol. Some concerns need to be addressed before being accepted for publication.
1. In all the IHC figures, please also provide figures with lower magnifications and negative control. And calculate the mean value of the positive staining, not just present the presentative figures. Besides, please double-check the scale bars in the figures and list the antibodies information in the method part.
2. For the flow cytometry data, please also provide the positive control and compensation methods
3. The confocal figures need to be updated. In Figure 3G, please add the DAPI stain. In Figures 6F and G, the nuclear stain is too dark for the 5uM and 10uM.
4. Please add method details for glucose and insulin tolerance tests. Please also explain the Figure 1F, why is the mice's blood glucose level so high, up to 800 mg/dL? I'm curious about the product name of the blood glucose monitor, which could read larger than 600 mg/dL.
Reviewer 2 Report
Park et al. investigate one possible mechanism for the progression of nonalcoholic fatty liver disease, specifically one of its types, nonalcoholic steatohepatitis (NASH). The role of platelets and erythrophagocytosis in the development of NASH is shown in an induced mouse model of NASH using a wide range of methods. The authors investigated the action of the well-known drug cilostazol, showing that it alleviates the course of the disease by preventing ectopic erythrophagocytosis and reducing iron overload-mediated liver ferroptosis. Separate experiments were carried out using human samples.
Notes:
1. 1. The STAMTM model used by the authors represents the development of NASH in diabetes. However, the article does not say anything about this and the results obtained are not discussed taking this factor into account. There is only a mention on page 2 (lines 49-51) “Increased hepatic fat and insulin resistance at the onset of NAFL represent a “first hit” and other “hits” are required for NAFL to progress into NASH”.
2. For me, the age of the mice used in the experiments for histological, biochemical, and other studies remained unclear. Because, on page 3, line 98, we read that “NASH was induced in 8-week-old C57BL/6 STAMTM mice”. But on page 12, lines 394-395 it says "Mice were sampled at 12 weeks of age".
3. How many animals were used in each experiment? Figure 2A says n=10 and page 14, line 488 says “There were 8 samples per group in the in vivo experiments”. What's right?
4. Page 3, lines 101-102. “The NAFLD activity score (NAS) developed by Kleiner et al. ….” It is necessary to cite the article and add it to the References section.
5. Page 4, lines 142–143. “NAS value determined by the deep learning module developed by Heinemann et al.” It is necessary to cite the article and add it to the References section.
6. Page 4, lines 123-124. “…HFD-induced hepatic steatosis…” HFD should be deciphered.
7. In the section "Materials and Methods" there is no description of the histological analysis, although the results obtained with its use are widely presented in the work.
8. There are repetitions in the "Discussion" section. For example:
- Page 11, lines 328-329. “HIO is an emerging issue commonly found in NAFLD and an exacerbating factor of NAFLD itself [14,15]. HIO is a frequent finding in the NAFLD population and is an aggravating factor for NAFLD.”
- On page 11 (lines 332-334) “Although the detailed mechanism of HIO is not yet clear, some reports point to increased erythrophagocytosis as one of the reasons for HIO ..” and on page 11, (lines 338-340) “ To date, the origin of iron in NASH liver remains unclear. Our findings suggest that hepatic erythrophagocytosis is an iron source in the livers of patients with NASH.”
9. Figure S1 shows the effects of cilostazol on the lipid and glucose profile. But by what methods and how this was done was not indicated in the article.
Minor editing of English language required
Reviewer 3 Report
The authors of this article present a study on the role of hepatic erythrophagocytosis and cilostazol in nonalcoholic fatty liver disease (NAFLD) and nonalcoholic steatohepatitis (NASH). They use human pathology samples, an in vivo mouse model, and an in vitro coculture system to investigate the effects of cilostazol on hepatic iron overload (HIO) and ferroptosis. They claim that cilostazol ameliorates hepatic steatosis and inflammation, inhibits platelet-mediated erythrophagocytosis, and attenuates ferroptosis in NASH models.
The article is well-written and concise, but it lacks some details and references that would support the authors’ claims. For example, the authors do not explain how cilostazol works as a potential therapeutic agent for NASH, what are the mechanisms of platelet activation and erythrophagocytosis in NASH liver, and how ferroptosis is related to HIO. The article would benefit from more citations of relevant literature and more evidence of the effects of cilostazol on NASH models (https://www.jci.org/articles/view/137468).
In addition to the above comments, I have some more queries/feedback to be addressed.
1. What is the rationale for choosing cilostazol as a potential therapeutic agent for NASH?
2. How did you measure the NAFLD activity score and the degree of hemolysis in STAMTM mice?
3. How did you isolate and activate THP-1 cells and RBCs for the insert-coculture system?
4. How did you quantify the erythrophagocytosis and ferroptosis of hepatocytes and THP-1 cells?
5. What are the limitations of your study and the implications of your findings for future research?
6. How did you determine the optimal dose and duration of cilostazol treatment for NASH mice? Did you observe any adverse effects or toxicity of cilostazol in your study?
7. Can author explain more in conclusion means expends the text.
8. The author should try to improve the quality of pictures in figures 3C and 5B.
The article is concise, clear, and well-written. After addressing the above questions, it should be accepted.
Round 2
Reviewer 1 Report
Thanks for the response. The authors addressed most of the concerns.
Reviewer 2 Report
Thank you. There are no comments.
Minor editing of English language required
Reviewer 3 Report
The Authors have addressed all of the reviewer's concerns with the original manuscript. The revised manuscript is ready for publication in IJMS.